# XX sex chromosome complement promotes atherosclerosis in mice

Yasir AlSiraj[1], Xuqi Chen[2], Sean E. Thatcher [1], Ryan E. Temel[3,4], Lei Cai[3,4], Eric Blalock[1], Wendy Katz [1], Heba M. Ali[1], Michael Petriello[5], Pan Deng [5], Andrew J. Morris [5], Xuping Wang[6,7,8], Aldons J. Lusis [6,7,8], Arthur P. Arnold[2], Karen Reue[8], Katherine Thompson[9], Patrick Tso[10] & Lisa A. Cassis[1]

Men and women differ in circulating lipids and coronary artery disease (CAD). While sex hormones such as estrogens decrease CAD risk, hormone replacement therapy increases risk. Biological sex is determined by sex hormones and chromosomes, but effects of sex chromosomes on circulating lipids and atherosclerosis are unknown. Here, we use mouse models to separate effects of sex chromosomes and hormones on atherosclerosis, circulating lipids and intestinal fat metabolism. We assess atherosclerosis in multiple models and experimental paradigms that distinguish effects of sex chromosomes, and male or female gonads. Pro-atherogenic lipids and atherosclerosis are greater in XX than XY mice, indicating a primary effect of sex chromosomes. Small intestine expression of enzymes involved in lipid absorption and chylomicron assembly are greater in XX male and female mice with higher intestinal lipids. Together, our results show that an XX sex chromosome complement promotes the bioavailability of dietary fat to accelerate atherosclerosis.

[1] Department of Pharmacology and Nutritional Sciences, University of Kentucky, Lexington 40536 KY, USA. [2] Integrative Biology and Physiology, University of California, Los Angeles 90095 CA, USA. [3] Saha Cardiovascular Research Center, University of Kentucky, Lexington 40536 KY, USA. [4] Department of Physiology, University of Kentucky, Lexington 40536 KY, USA. [5] Division of Cardiovascular Medicine, Department of Internal Medicine, University of Kentucky, and Lexington Veterans Affairs Medical Center, Lexington 40536 KY, USA. [6] Department of Microbiology, Immunology & Molecular Genetics, Human Genetics, College of Medicine, University of California, Los Angeles 90095 CA, USA. [7] Microbiology, Immunology and Molecular Genetics, University of California, Los Angeles 90095 CA, USA. [8] Human Genetics, University of California, Los Angeles 90095 CA, USA. [9] Department of Statistics, University of Kentucky, Lexington 40536 KY, USA. [10] Department of Pathology, University of Cincinnati, Cincinnati 45215 OH, USA. Correspondence and requests for materials should be addressed to L.A.C. (email: lcassis@uky.edu)

Sex chromosomes and sex hormones are the primary determinants of biological sex. A plethora of research has focused on the role of sex hormones as mediators of sex differences in a variety of diseases, most especially cardiovascular diseases[1,2]. Generally, results from these studies suggest that estrogens have beneficial effects on circulating lipid profiles (e.g., increase HDL)[3–6] and protect against coronary artery disease (CAD)[7–10], and that these benefits are typically lost in post-menopausal females. Notably, some studies report that post-menopausal females exhibit a pro-atherogenic lipid profile and an increase in CAD to a level that not only catches up to, but exceeds that of age-matched males[11–13]. This suggests that female gonadal hormones, such as estrogens, are unlikely to be the only determinant of sex differences in CAD risk.

In comparison to sex hormones, genes residing on sex chromosomes have been relatively under-studied as causes of sex differences in disease development. While the Y chromosome has evolved to contain few genes, the X chromosome contains as much as 5% of the human genome, and could thus potentially mediate sex differences in a variety of factors and/or diseases[14,15]. Unfortunately, many large-scale genome-wide association studies (GWAS)[16], including GWAS studies in subjects with CAD[17,18], have neglected analysis of genes residing on sex chromosomes. Thus, the contribution of sex chromosome genes to CAD, and other common diseases, is not well characterized.

We use the Four Core Genotypes (FCG) mouse model[19–21], which generates XX and XY female mice with ovaries, and XX and XY male mice with testes, to define the role of sex chromosome genotype on circulating lipids and atherosclerosis. Our results demonstrate that an XX sex chromosome genotype, relative to XY, promotes the development of atherosclerotic lesions in multiple mouse models and this is associated with profound dyslipidemia, enhanced adiposity, and augmented dietary fat bioavailability. While gonadal hormones also regulated some of these factors, the pronounced effects of XX sex chromosome genotype persist in gonadectomized (GDX) mice. Moreover, higher serum lipids and atherosclerosis are evident in XX female and male mice under different experimental paradigms (e.g., diet, genetic background, gonadectomy), and multiple linear regression analysis reveals sex chromosome genotype as an explanatory variable for the development of atherosclerosis. Our data suggest that the greater atherosclerosis susceptibility in XX compared to XY mice is associated with enhanced absorption and bioavailability of dietary fat, which likely influences serum lipid levels and adiposity. Our findings may have important ramifications for human health, particularly following menopause, when protective effects of female sex hormones are lost, and the effects of an XX sex chromosome genotype may contribute to pro-atherogenic lipid profiles and CAD.

## Results

### XX males and females have high food intake and body weight.
We generated FCG mice (XX females, XX males, XY males, and XY females) on an $Ldlr^{-/-}$ C57BL/6 J background to study effects of sex chromosome genotype on serum lipids and atherosclerosis. Mice were defined as male or female on the presence of testes or ovaries, respectively. Serum testosterone concentrations were higher in male than female mice, regardless of sex chromosome genotype (Male, XX: 2.89 ± 1.28, XY: 1.65 ± 0.85; Female, XX: 0.55 ± 0.08, XY: 0.33 ± 0.14 ng/ml; $P < 0.005$ for male compared to female by 2-way ANOVA with Holm–Sidak test). However, there were no differences in serum testosterone concentrations between XX and XY mice, regardless of sex. Gonadectomy decreased significantly serum testosterone concentrations in both sexes, with more pronounced reductions in males (GDX, Male, XX:

0.24 ± 0.04, XY: 0.15 ± 0.06; Female, XX: 0.24 ± 0.09, XY: 0.06 ± 0.04 ng/ml; $P < 0.001$ compared to intact within sex and sex chromosome genotype by 2-way ANOVA with Holm–Sidak test). We were unable to quantify serum estrogen concentrations because of interference from plasma lipids within the ELISA. At baseline and following 1 week of the Western diet, males had significantly higher body weights than females, regardless of sex chromosome genotype (Fig. 1a; $P < 0.001$ 3-way ANOVA with Holm–Sidak test). Moreover, XX mice (at baseline or following 1 week of Western diet), regardless of whether they were females or males, had significantly higher body weights (Fig. 1a; $P < 0.001$, 3-way ANOVA with Holm–Sidak test) and lean mass (Fig. 1b; $P < 0.001$, 3-way ANOVA with Holm–Sidak test) than XY mice of either sex, in agreement with previous reports[22]. Fat mass was significantly higher at baseline in XX male ($P < 0.001$, 3-way ANOVA with Holm–Sidak test), but not XX female mice ($P = 0.383$, 3-way ANOVA with Holm–Sidak test) compared to XY mice of either sex (Fig. 1c; $P < 0.001$, 3-way ANOVA with Holm–Sidak test).

Female mice had significant increases in food intake and activity compared to males, regardless of sex chromosome genotype, while males of each genotype had higher energy expenditure than females (Fig. 1d–f). Moreover, XX mice with higher body weights and fat mass had significantly higher food intake (male and female, Fig. 1d; $P = 0.028$, 2-way ANOVA with Holm–Sidak test), activity (female, Fig. 1e; $P = 0.033$, 2-way ANOVA with Holm–Sidak test), and energy expenditure (male and female, Fig. 1f; $P < 0.001$, 2-way ANOVA with Holm–Sidak test) than XY mice of either sex. When male and female mice were challenged short-term for 1 week with a Western diet, differences in body weight (Fig. 1a), lean and fat mass (Fig. 1b, c), food intake (Fig. 1g; $P = 0.04$, 2-way ANOVA with Holm–Sidak test), activity (female, Fig. 1h; $P = 0.03$, 2-way ANOVA with Holm–Sidak test) and energy expenditure (Fig. 1i; $P = 0.003$, 2-way ANOVA with Holm–Sidak test) of XX compared to XY mice (male or female) were augmented.

### XX males and females have high lipids and atherosclerosis.
We fed male and female XX and XY $Ldlr^{-/-}$ mice a Western diet for 4 months to examine effects of sex chromosome genotype on serum lipids and atherosclerosis. To separate the contribution of sex chromosomes and sex hormones, male and female mice of each sex chromosome genotype were either gonadally intact (Intact) or surgically gonadectomized (GDX) two weeks prior to initiation of the Western diet. Males (Intact) had increased body weights (Fig. 2a; $P < 0.001$, 3-way ANOVA with Holm–Sidak test) and fat mass (Fig. 2b, $P < 0.001$, 3-way ANOVA with Holm–Sidak test) compared to females, regardless of sex chromosome genotype. Moreover, XX female and male mice (Intact) had greater body weights (Fig. 2a, b; $P < 0.001$, 3-way ANOVA with Holm–Sidak test), with 1.4–2.7-fold increases of the weights of white adipose tissue (Table 1) compared to XY mice of either sex. Gonadectomy decreased body weights of male, but not female $Ldlr^{-/-}$ mice, regardless of sex chromosome genotype (Fig. 2a; $P < 0.001$, 3-way ANOVA with Holm–Sidak test). Moreover, greater body weights (Fig. 2a, b) and fat mass (Table 1) of XX mice, relative to XY, persisted after gonadectomy.

Lipid content of serum from XX $Ldlr^{-/-}$ mice fed a Western diet was visibly greater than XY $Ldlr^{-/-}$ mice (Fig. 2c). Serum total cholesterol concentrations were significantly higher in male compared to female mice, regardless of sex chromosome genotype or surgery (GDX) (Fig. 2d; $P < 0.001$, 3-way ANOVA with Holm–Sidak test). Moreover, serum total cholesterol concentrations were markedly higher ( > 3-fold) in XX female and male mice compared to XY mice of either sex (Fig. 2d; $P <$

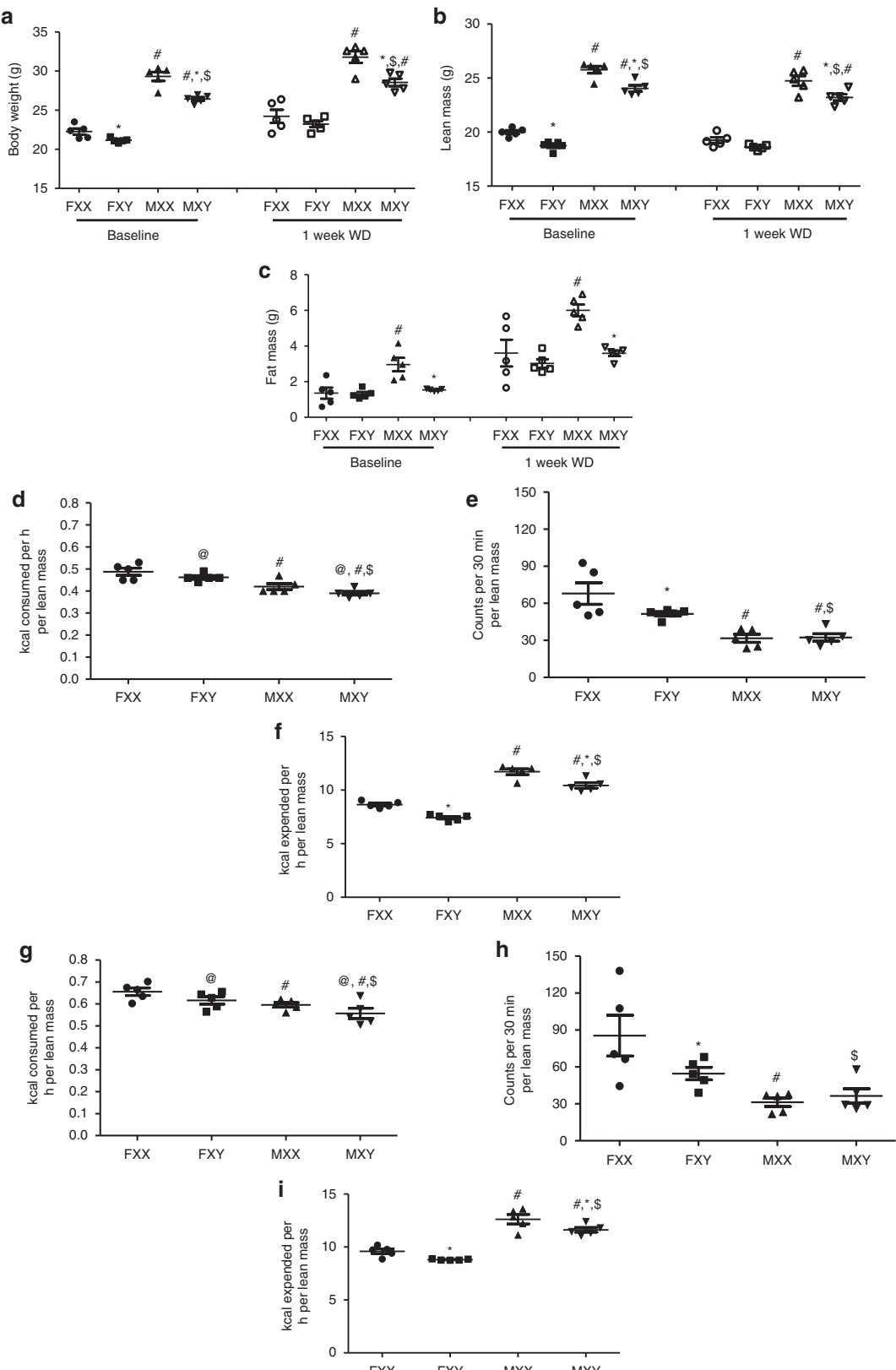

0.001, 3-way ANOVA with Holm–Sidak test), and these effects persisted in GDX mice. However, there was no significant effect of gonadectomy of male or female $Ldlr^{-/-}$ mice on serum cholesterol concentrations, regardless of sex chromosome genotype. Serum concentrations of VLDL- (Fig. 2e; $P < 0.001$, 3-way ANOVA with Holm–Sidak test) and LDL-cholesterol (Fig. 2f; $P <$

0.001, 3-way ANOVA with Holm–Sidak test) were also higher in male compared to female mice, regardless of sex chromosome genotype or surgery. Notably, serum VLDL- and LDL-cholesterol concentrations were also markedly higher in XX than XY mice, regardless of sex or surgery. There was no difference in serum HDL-cholesterol concentrations between males and females,

**Fig. 1** XX male and female mice have higher body weight, fat mass and food intake. **a** Body weight of mice of each sex chromosome complement and gonadal sex at baseline (when fed standard murine diet) or after 1 week of consumption of a Western diet (WD). Lean (**b**) and fat (**c**) mass (gm). **d** Food intake, normalized to lean mass, of mice fed standard murine diet. **e** Physical activity, normalized to lean mass, of mice fed standard murine diet. **f** Energy expenditure, normalized to lean mass, of mice fed standard murine diet. **g** Food intake, normalized to lean mass, of mice fed a Western diet (1 week). **h** Physical activity, normalized to lean mass, of mice fed a Western diet. **i** Energy expenditure, normalized to lean mass, of mice fed a Western diet. Symbols represent individual mice per group ($n = 5$ mice/group) per measurement, with horizontal lines representing mean ± SEM. *$P < 0.05$ compared to XX within gonadal sex. #$P < 0.05$ compared to female within sex chromosome complement. $$P < 0.05$ compared to XX females. @$P < 0.05$ main effect of sex chromosome complement. Data were analyzed by 3-way ANOVA (A-C) with Holm–Sidak test, or by 2-way ANOVA (**d–i**) with Holm–Sidak test. Source data are available as a Source Data file

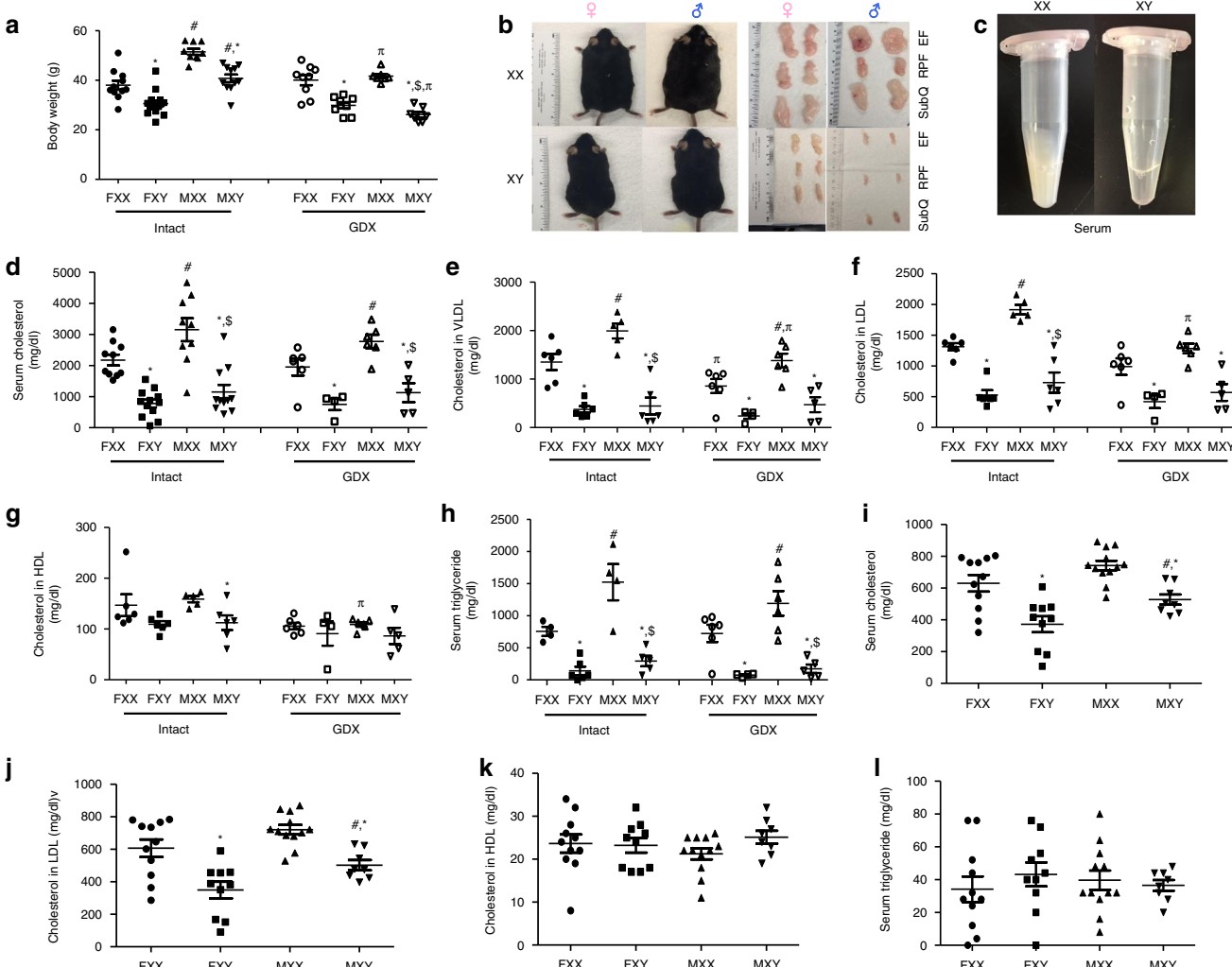

**Fig. 2** An XX sex chromosome complement promotes obesity and dyslipidemias. **a** Body weights (gm) of male and female intact and gonadectomized (GDX) mice of each genotype (Intact: FXX, $n = 11$; FXY, $n = 14$; MXX, $n = 9$; MXY, $n = 11$; GDX: FXX, $n = 9$; FXY, $n = 8$; MXX, $n = 7$; MXY, $n = 7$). **b** Representative pictures of mice from each group (left), with adipose tissue depots illustrated. **c** Representative pictures of serum from XX and XY male mice. **d** Total serum cholesterol concentrations (Intact: FXX, $n = 10$; FXY, $n = 12$; MXX, $n = 9$; MXY, $n = 11$; GDX: FXX, $n = 6$; FXY, $n = 4$; MXX, $n = 6$; MXY, $n = 5$). Concentrations of very low density lipoprotein (VLDL)-cholesterol (**e**) (Intact: FXX, $n = 6$; FXY, $n = 6$; MXX, $n = 5$; MXY, $n = 6$; GDX: FXX, $n = 6$; FXY, $n = 4$; MXX, $n = 6$; MXY, $n = 5$), low density lipoprotein (LDL)-cholesterol (**f**) (Intact: FXX, $n = 6$; FXY, $n = 6$; MXX, $n = 5$; MXY, $n = 6$; GDX: FXX, $n = 6$; FXY, $n = 4$; MXX, $n = 6$; MXY, $n = 5$) and high density lipoprotein (HDL)-cholesterol (**g**) (Intact: FXX, $n = 6$; FXY, $n = 4$; MXX, $n = 6$; MXY, $n = 5$; GDX: FXX, $n = 6$; FXY, $n = 4$; MXX, $n = 6$; MXY, $n = 5$) and TG (**h**) (Intact: FXX, $n = 4$; FXY, $n = 6$; MXX, $n = 4$; MXY, $n = 5$; GDX: FXX, $n = 6$; FXY, $n = 4$; MXX, $n = 6$; MXY, $n = 5$) in $Ldlr^{-/-}$ mice fed a Western diet for 4 months. Concentrations of total serum cholesterol (**i**) (FXX, $n = 11$; FXY, $n = 10$; MXX, $n = 12$; MXY, $n = 8$), LDL-cholesterol (**j**) (FXX, $n = 11$; FXY, $n = 10$; MXX, $n = 12$; MXY, $n = 8$), HDL-cholesterol (**k**) (FXX, $n = 11$; FXY, $n = 10$; MXX, $n = 12$; MXY, $n = 8$) and TG (**l**) (FXX, $n = 11$; FXY, $n = 10$; MXX, $n = 12$; MXY, $n = 8$) in $Apoe^{-/-}$ mice fed a standard murine diet. Symbols represent individual mice per group per measurement, with horizontal lines representing mean ± SEM. *$P < 0.05$ compared to XX within gonadal sex. #$P < 0.05$ compared to female within sex chromosome complement. $$P < 0.05$ compared to XX females. Data were analyzed by 3-way ANOVA (**a**, **d–h**) with Holm–Sidak test, or by 2-way ANOVA (**i–l**) with Holm–Sidak test. Source data are available as a Source Data file. RPF retroperitoneal, EF epididymal, SubQ subcutaneous

**Table 1 Characteristics of mice of each group**

| | Parameter | FXX | FXY | MXX | MXY |
|---|---|---|---|---|---|
| Intact $Ldlr^{-/-}$ on Western Diet | Retroperitoneal Fat/body weight (%) | 2.04 ± 0.3 | 0.92 ± 0.2 | 2.54 ± 0.17 | 1.57 ± 0.17 |
| | SubQ Fat/body weight (%) | 2.19 ± 0.17 | 1.85 ± 0.17* | 3.5 ± 0.27 | 1.89 ± 0.24* |
| | Gonadal Fat/body weight (%) | 6.06 ± 0.65 | 3.06 ± 0.4* | 4.94 ± 0.22# | 4.59 ± 0.38* |
| | Liver/body weight (%) | 5.8 ± 0.23 | 6.3 ± 0.11 | 7.9 ± 0.24 | 5.8 ± 0.17 |
| Gonadectomized $Ldlr^{-/-}$ on Western Diet | Retroperitoneal Fat/body weight (%) | 3 ± 0.34 | 1.2 ± 0.2 | 2.1 ± 0.12 | 0.59 ± 0.09 |
| | SubQ Fat/body weight (%) | 3.0 ± 0.21 | 1.7 ± 0.2* | 3.66 ± 0.43 | 1.27 ± 0.15* |
| | Gonadal Fat/body weight (%) | 5.1 ± 0.23 | 2.5 ± 0.3* | 5.18 ± 0.62 | 2.01 ± 0.45* |
| | Liver/body weight (%) | 6.3 ± 0.27 | 5.9 ± 0.1 | 5.88 ± 0.25 | 5.3 ± 0.17 |
| $Apoe^{-/-}$ on Chow Diet | SubQ Fat/body weight (%) | 0.87 ± 0.13 | 0.60 ± 0.10 | 0.85 ± 0.10 | 0.73 ± 0.05 |
| | Gonadal Fat/body weight (%) | 0.86 ± 0.23 | 0.40 ± 0.13 | 0.73 ± 0.10 | 0.55 ± 0.08 |
| | Liver/body weight (%) | 4.44 ± 0.17 | 4.70 ± 0.22 | 4.30 ± 0.10 | 4.42 ± 0.18 |
| C57BL/6 J on Atherogenic Diet | SubQ Fat/body weight (%) | 0.95 ± 0.12 | 0.70 ± 0.06* | 0.64 ± 0.05 | 0.47 ± 0.03 |
| | Gonadal Fat/body weight (%) | 2.16 ± 0.31 | 1.11 ± 0.16* | 1.33 ± 0.23 | 0.30 ± 0.16 |
| | Liver/body weight (%) | 7.50 ± 0.40 | 9.51 ± 0.35* | 8.45 ± 0.51 | 11.6 ± 1.0* |

Data are mean ± SEM
Data are analyzed by 2-way ANOVA with Holm–Sidak test
*$P < 0.05$ compared to XX within gonadal sex. #$P < 0.05$ compared to female within genotype

regardless of surgery (Fig. 2g). Serum HDL-cholesterol concentrations were higher in XX males, but not XX females compared to XY mice (Fig. 2g; $P = 0.018$, 3-way ANOVA with Holm–Sidak test). This difference in serum HDL-cholesterol concentrations between male XX and XY mice was not present in GDX mice. Serum triglyceride (TG) concentrations were also higher in male than female mice, regardless of sex chromosome genotype (Fig. 2h; $P < 0.001$, 3-way ANOVA with Holm–Sidak test). Moreover, XX mice had markedly higher serum TG concentrations than XY mice, and this effect persisted in GDX mice (Fig. 2h; $P < 0.001$, 3-way ANOVA with Holm–Sidak test). There was no effect of GDX on serum TG concentrations in male or female mice of either sex chromosome genotype.

To determine if XX effects on serum lipids and atherosclerosis were specific to an $Ldlr^{-/-}$ background and/or required a Western diet, we examined serum cholesterol, TG, LDL- and HDL-cholesterol concentrations in FCG mice fed standard murine diet for 4 months but made hypercholesterolemic by apolipoprotein E deficiency ($Apoe^{-/-}$). To remove influences of sex hormones, these studies were performed in GDX mice. Similar to findings from $Ldlr^{-/-}$ mice, serum total cholesterol and LDL-cholesterol concentrations were higher in male compared to female $Apoe^{-/-}$ mice, regardless of sex chromosome genotype (Fig. 2i, j; $P = 0.004$, 3-way ANOVA with Holm–Sidak test). Moreover, XX male and female $Apoe^{-/-}$ mice had higher serum total cholesterol (Fig. 2i; $P < 0.001$, 2-way ANOVA with Holm–Sidak test) and LDL-cholesterol (Fig. 2j; $P < 0.001$, 2-way ANOVA with Holm–Sidak test) concentrations than XY mice of either sex. However, neither serum HDL-cholesterol (Fig. 2k; $P = 0.32$, 2-way ANOVA with Holm–Sidak test) nor TG concentrations (Fig. 2l; $P = 0.66$, 2-way ANOVA with Holm–Sidak test) were different among the four genotypes.

We assessed the influence of sex chromosome complement on atherosclerosis in three independent FCG mouse genetic backgrounds: $Ldlr^{-/-}$ mice (Intact and GDX) fed a Western diet, GDX $Apoe^{-/-}$ mice fed standard murine diet, and C57BL/6 J mice fed a cholesterol-enriched atherogenic diet. In aortic arches of $Ldlr^{-/-}$ mice, the percent of intimal surface area covered by atherosclerotic lesions was significantly greater in male than female mice (Intact), regardless of sex chromosome genotype (Fig. 3a; $P = 0.029$, 3-way ANOVA with Holm–Sidak test). Female XX, but not male XX mice (Intact) had more atherosclerosis than XY mice of the respective sex (Fig. 3a, b; $P = 0.01$, 3-way ANOVA with Holm–Sidak test). Gonadectomy increased atherosclerosis in female XX, but not female XY mice

(Fig. 3a, b; $P = 0.002$, 3-way ANOVA with Holm–Sidak test), suggesting a protective role for female gonadal hormones that requires an XX sex chromosome genotype. In contrast, male XY, but not male XX mice exhibited significantly less atherosclerosis in the GDX groups relative to intact XY males (Fig. 3a, b; $P = 0.046$, 3-way ANOVA with Holm–Sidak test), suggesting interactions between testicular hormones and an XY sex chromosome genotype on lesion development. Following gonadectomy, atherosclerosis of XX mice was markedly greater than XY mice, regardless of sex (Fig. 3a, b; $P < 0.001$, 3-way ANOVA with Holm–Sidak test), demonstrating a robust effect of sex chromosome genotype.

We also quantified atherosclerosis in aortic sinus tissue sections of FCG $Ldlr^{-/-}$ mice, where lesion areas were not significantly different between male and female mice, regardless of sex chromosome genotype or surgery (Fig. 3c). However, XX male and female mice had significantly greater atherosclerotic lesion areas compared to XY mice of either sex (Fig. 3c, d; $P < 0.001$, 3-way ANOVA with Holm–Sidak test), which persisted in GDX mice. In aortic sinus tissue sections from GDX $Apoe^{-/-}$ (Fig. 3e, f) fed with standard murine diet or GDX C57BL/6 J FCG mice fed with a HF diet for 4 months, there were no differences in atherosclerotic lesion areas between male and female mice, regardless of sex chromosome genotype (Fig. 3e). However, similar to $Ldlr^{-/-}$ FCG mice, XX female and male GDX $Apoe^{-/-}$ FCG mice (Fig. 3e, f; $P < 0.0001$, 2-way ANOVA with Holm–Sidak test), as well as XX female and male GDX C57BL/6 J FCG mice fed with an atherogenic diet for 4 months (Fig. 3g; $P = 0.001$, 2-way ANOVA with Holm–Sidak test) had significantly greater aortic sinus lesion areas compared to XY mice of either sex. These results demonstrate the robust effect of the XX genotype on atherosclerosis under three complementary experimental paradigms.

A variety of parameters quantified in these studies could contribute to higher levels of atherosclerosis in XX compared to XY mice, including higher energy intake, body weight, fat mass or differences based on gonadal sex, sex chromosome genotype, or genetic background (e.g., C57BL/6 J vs. $Ldlr^{-/-}$ mice). We used a multiple linear regression model with log-transformed atherosclerotic lesion area in aortic sinus as the response variable, and examined the relationship of each of the above described explanatory variables within the model to determine their relationship to atherosclerosis. For this model, reference groups for the analysis were $Apoe^{-/-}$ mice (genetic background), GDX (surgery), females (sex), and XX sex chromosome genotype (sex

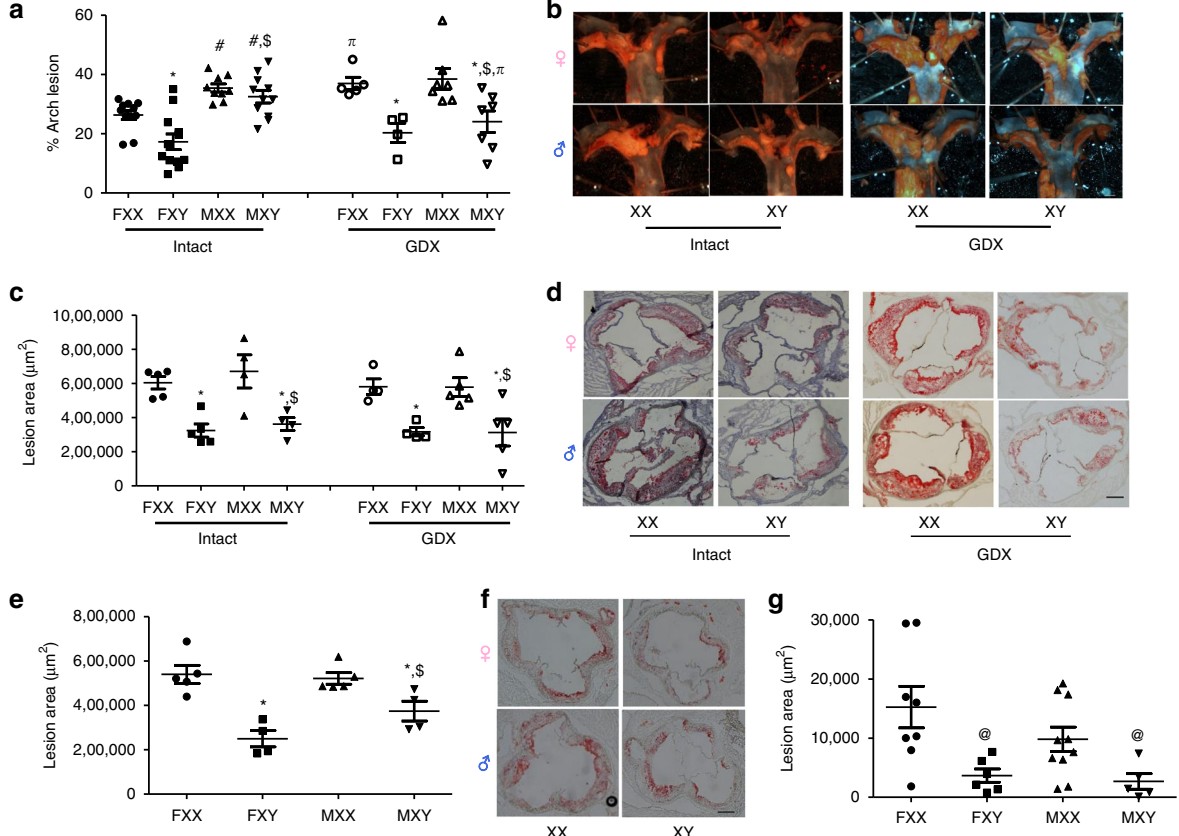

**Fig. 3** An XX sex chromosome complement augments atherosclerosis in male and female mice. **a** Atherosclerotic lesion surface area, expressed as a percentage of the aortic arch, in aortic arches from male and female XX and XY intact or gonadectomized (GDX) $Ldlr^{-/-}$ mice. (Intact: FXX, $n = 11$; FXY, $n = 12$; MXX, $n = 9$; MXY, $n = 11$; GDX: FXX, $n = 5$; FXY, $n = 4$; MXX, $n = 7$; MXY, $n = 7$). **b** Representative aortic arch, stained with Oil Red O, from $Ldlr^{-/-}$ mice of each group. **c** Atherosclerotic lesion area, quantified as Oil Red O staining, in tissue sections from the aortic sinus of male and female XX and XY intact or GDX $Ldlr^{-/-}$ mice. (Intact: FXX, $n = 4$; FXY, $n = 5$; MXX, $n = 4$; MXY, $n = 4$; GDX: FXX, $n = 4$; FXY, $n = 4$; MXX, $n = 5$; MXY, $n = 5$). **d** Representative aortic sinus tissue sections, stained with Oil Red O, from $Ldlr^{-/-}$ mice of each group. **e** Atherosclerotic lesion area in tissue sections from XX and XY male and female $Apoe^{-/-}$ GDX mice. (FXX, $n = 5$; FXY, $n = 4$; MXX, $n = 5$; MXY, $n = 4$). **f** Representative aortic sinus tissue sections from $Apoe^{-/-}$ GDX mice of each group. **g** Atherosclerotic lesion area in tissue sections from aortic sinus of XX and XY male and female C57BL/6 GDX mice fed an atherogenic diet. Symbols represent individual mice per group per measurement, with horizontal lines representing mean ± SEM. (FXX, $n = 8$; FXY, $n = 6$; MXX, $n = 10$; MXY, $n = 5$). *$P < 0.05$ compared to XX within gonadal sex. #$P < 0.05$ compared to female within sex chromosome complement. $$P < 0.05$ compared to XX females. Data were analyzed by 3-way ANOVA (A,C) with Holm–Sidak test, or by 2-way ANOVA (**e**, **g**) with Holm–Sidak test. Scale bar $= 200\,\mu m$. Source data are available as a Source Data file

**Table 2 Multiple linear regression of explanatory variables to atherosclerotic lesion area within the aortic sinus of FCG mice**

|  | Estimate | Standard Error | t-value | P-value |
|---|---|---|---|---|
| (Intercept) | 13.23 | 0.49 | 26.8 | <0.0000000000000002 |
| Body weight (g) | 0.01 | 0.02 | 0.56 | 0.58 |
| Sera cholesterol (mg/dl) | −0.00 | 0.00 | −1.43 | 0.16 |
| Gonadal fat (%) | 0.08 | 0.09 | 0.88 | 0.38 |
| *Genetic background*: |  |  |  |  |
| $Ldlr^{-/-}$ mice | −0.07 | 0.36 | −0.19 | 0.85 < 0.0000000000000002 |
| C57BL/6 mice | −4.62 | 0.23 | −19.95 | 0.85 < 0.0000000000000002 |
| Sex | −0.09 | 0.17 | −0.56 | 0.58 |
| Surgery (Intact) | 0.01 | 0.27 | 0.03 | 0.98 |
| Sex chromosome genotype | −0.99 | 0.21 | −4.77 | 0.000011 |

Reference groups were $Apoe^{-/-}$ mice (genetic background), female (sex), GDX (surgery), and XX genotype (sex chromosome genotype)

chromosome genotype) (Table 2). We included all mice in the analysis for which we had measurements on all variables. After adjusting for all other variables, there were two explanatory variables that were significant for the development of atherosclerosis, namely sex chromosome genotype and genetic background (Table 2).

**Livers of XX males and females have diverse gene expression.** Elevations in serum lipids of XX mice could result from alterations in cholesterol and/or lipid homeostasis in liver, a major organ for lipoprotein synthesis, secretion and clearance. To focus on sex chromosome influences on transcriptional profiles in the absence of sex hormones, livers from GDX mice were used. We

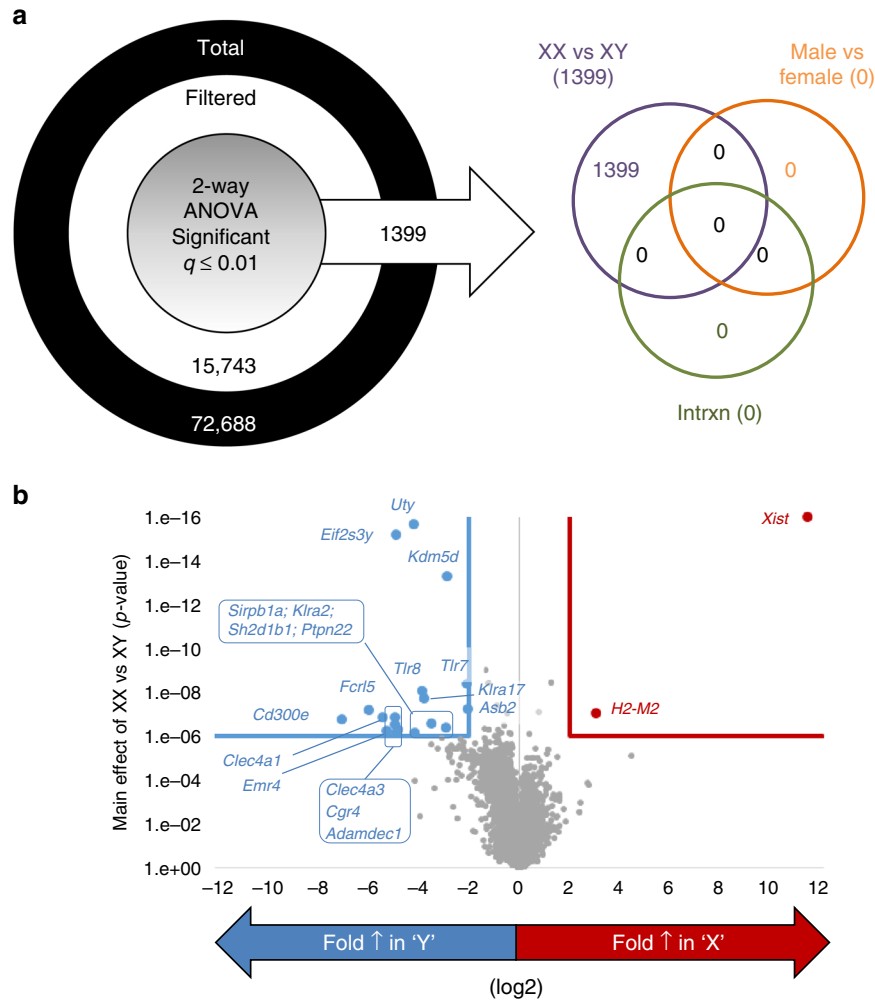

**Fig. 4** Sex chromosome complement influences hepatic gene expression of $Ldlr^{-/-}$ gonadectomized (GDX) mice. **a** Total probe sets on the array were filtered to retain annotated transcript clusters with reliable signal intensity. The filtered data set was analyzed by two-way ANOVA. The number of genes whose expression was significantly changed (multiple testing corrected $q$ value ≤ 0.01) by the main effects of gene status (XX vs XY), biological sex (male vs. female), as well as by interaction, are listed. Note that only genes significant by the chromosome effect survived multiple testing correction. **b** Fold change in gene expression (log 2 scale, x-axis) and statistical significance (p value, y-axis) for the main effect of chromosome are plotted. Genes labelled in blue exhibited significant increases in livers from XY compared to XX mice; genes labelled in red exhibited significant increases in livers from XX compared to XY mice. (FXX, $n = 5$; FXY, $n = 5$; MXX, $n = 4$; MXY, $n = 5$). Source data are available as a Source Data file

performed transcriptome analysis on livers from GDX XX and XY male and female $Ldlr^{-/-}$ mice after 4 months of Western diet using Affymetrix Mouse Transcriptome Assay 1.0 assays. There was no major effect of sex on liver gene expression (Fig. 4a, Male vs. Female). However, a total of 1,399 genes exhibited highly significant differences (2-way ANOVA, False Discovery Rate < 0.01; Fig. 4a, b, Supplementary Data 1) by sex chromosome genotype (XX or XY). Volcano plots of the sex chromosome effect with highly stringent cutoffs ( > 4-fold change, $P < 1 \times 10^{-6}$) demonstrated that the expression of genes on sex chromosomes was strongly influenced (Fig. 4b). As expected, genes within the male-specific region of the Y chromosome (e.g., *Uty, Kdm5d, Eif2s3y*) were significantly greater in XY livers (Fig. 4b), while *Xist* (known to be expressed only in XX cells) was significantly greater in XX livers. Biological pathway analyses revealed that a large number of genes involved in the immune response (197) differed in livers from XY compared to XX male and female mice (Table 3). We did not observe an effect of sex chromosome genotype on pathways related to hepatic cholesterol synthesis

| Table 3 XY vs XX liver transcriptional pathway over-representation | | | |
|---|---|---|---|
| **G.O.** | **Description** | **#** | **P value** |
| 0006955 | Immune response | 197 | 7.62E-52 |
| 0030029 | Actin filament-based process | 95 | 3.51E-21 |
| 0006897 | Endocytosis | 82 | 2.33E-20 |
| 0007264 | Small GTPase mediated signal transduction | 72 | 7.10E-16 |
| 0005925 | Focal adhesion | 60 | 7.68E-13 |
| 0018212 | Peptide-tyrosine modification | 46 | 1.18E-12 |
| 0001944 | Vasculature development | 68 | 4.43E-08 |
| 0072593 | Reactive oxygen species metabolic process | 33 | 1.58E-07 |

Column titles: G.O. (Gene Ontology accession identifier); #- number of genes found significant with category; p value- modified Fisher's Exact Test using #−1 (EASE score)

(Table 3), although some individual cholesterol-related genes were different between genotypes (Supplementary Data 1).

Since serum TG and cholesterol concentrations were greater in XX female or male mice compared to XY mice, we quantified hepatic TG and cholesterol concentrations, and examined gross morphology of liver tissue. Moreover, since higher serum lipids were present in GDX XX compared to XY males and females, indicating a primary effect of sex chromosome genotype, we focused on livers from GDX mice. Hepatic TG and cholesterol contents were greater in XX than XY females, but not in livers from XX vs. XY males (Supplementary Figure 1A, B; $P = 0.002$, 2-way ANOVA with Holm–Sidak test). Tissue sections from livers of XX and XY male and female mice fed the Western diet for 4 months had similar gross morphology (Supplementary Figure 1C). Elevations in hepatic cholesterol concentrations could arise from increased synthesis or decreased cholesterol secretion. As transcriptome analysis did not identify cholesterol home-ostasis pathways as different between livers from XX and XY mice, we quantified hepatic TG secretion as a major determinant of serum cholesterol concentrations in male and female XX and XY $Ldlr^{-/-}$ mice fed standard murine diet. Plasma TG concentrations over time in fasted XX and XY male and female mice (after injection with poloxamer to inhibit lipoprotein lipase-mediated lipolysis, Supplementary Figure 2A) and TG secretion rates (Supplementary Figure 2B) were higher in males compared to females, but there was no significant effect of sex chromosome genotype. Newly synthesized apolipoprotein B48 levels were higher in male XX and XY mice compared to female mice of each sex chromosome genotype ($P < 0.05$), but there was no difference between genotypes (Supplementary Figure 2C, D). In contrast, newly synthesized apolipoprotein B100 levels were higher in XX than XY mice, regardless of sex (Supplementary Figure 2C,E; $P = 0.029$, 2-way ANOVA with Holm–Sidak test), consistent with higher LDL/VLDL cholesterol concentrations of XX mice.

**XX male and female intestines have augmented lipid handling.** The intestinal tract absorbs dietary fat and is an important determinant of circulating lipids. Thus, we quantified mRNA abundance of a variety of genes (cluster of differentiation 36 (Cd36), fatty acid binding protein 1 (Fabp1), fatty acid binding protein 2 (Fabp2), secretion associated Ras related GTPase 1B (Sar1b), apolipoprotein B (Apob), diacylglycerol acyltransferases (Dgat1 and Dgat2), monoacylglycerol O-acyltransferase 2 (Mogat2), and microsomal triglyceride transfer protein (Mttp)) involved in the absorption and synthesis of dietary fat in small intestines from XX and XY GDX male and female FCG $Ldlr^{-/-}$ mice fed the Western diet for 4 months. mRNA abundance of Cd36, Fabpt1, Fabp2, Mogat2, Sar1b, Dgat1 and Apob was not different between males and females, and was not influenced by sex chromosome genotype (Supplementary Figure 3; $P > 0.05$, 2-way ANOVA with Holm–Sidak test). Similarly, mRNA abundance of Dgat2, an enzyme that produces TG from absorbed fatty acids[23,24] (Fig. 5a; $P = 0.008$, 2-way ANOVA with Holm–Sidak test), and Mttp (Fig. 5b; $P = 0.001$, 2-way ANOVA with Holm–Sidak test), which assembles TG into chylomicrons[25–27] was not different in small intestines of GDX male compared to female mice, regardless of sex chromosome genotype. However, mRNA abundance of both Dgat2 and Mttp was higher in intestines from XX compared to XY mice of either sex. In support of greater expression of these genes in small intestines of XX mice, intestinal TG content was also higher in female, but not male XX compared to XY mice (Fig. 5c; $P = 0.023$, 2-way ANOVA with Holm–Sidak test). Analysis of intestinal lipids by HPLC/mass spectrometry demonstrated that while there were no overall dif-ferences between males and females, there were greater palmitic

acid/palmitic acid/oleic acid-containing lipids in small intestines of XX compared to XY females, but not males (Fig. 5d; $P = 0.034$, 2-way ANOVA with Holm–Sidak test). Several other inter-mediates of TG synthesis showed similar patterns with higher levels of fatty acids in XX than XY small intestines (Supple-mentary Figure 4). To determine if these differences influenced dietary lipid absorption, we quantified fat absorption in female and male XX and XY $Ldlr^{-/-}$ mice fed a semisynthetic diet containing dietary fat and 5% sucrose polybehenate as a non-absorbable polyester that has physical properties of dietary fat[28]. There were no differences in % fat absorption between male and female mice, regardless of sex chromosome genotype (Fig. 5f). However, there was a trend for higher fat absorption in XX female and male mice compared to XY mice of either sex (Fig. 5e; $P = 0.067$, Kruskal–Wallis ANOVA on rank).

Finally, with the growing appreciation that the gut microbiota may play a role in cholesterol homeostasis and host health, we characterized overall microbiota differences utilizing 16 S rRNA sequencing and alpha diversity measurements from FCG female and male XX and XY $Ldlr^{-/-}$ mice fed the Western diet for 4 weeks. Female mice exhibited significantly higher alpha diversity as determined by Chao1, Phylogenetic Diversity (PD) whole tree, and observed Operational taxonomic units (OTU) indices (Supplementary Figure 5A-C; $P = 0.037, 0.019, 0.028$, respectively) than males, regardless of sex chromosome genotype. However, while there was an overall trend for higher diversity in XY compared to XX mice, these effects were not significant ($P = 0.11, 0.11, 0.086$). No differences were observed using the calculated Shannon diversity index (Supplementary Figure 5D).

## Discussion
Our findings shed light on causes of sex differences in common cardiovascular diseases such as CAD. We demonstrate that relative to XY, mice with an XX sex chromosome genotype exhibit the following: (1) markedly elevated serum cholesterol and TG concentrations, effects that were found in different experimental paradigms (e.g., diet, genetic background, gona-dectomy), (2) profound elevations in atherosclerosis, (3) altered expression of hepatic genes associated with immune pathways, (4) similar hepatic TG secretion but higher levels of newly syn-thesized apolipoprotein B100, (5) greater expression levels of key genes in small intestine involved in lipid absorption and chylo-micron assembly, higher intestinal lipid content, and modest elevations in % fat absorption (Fig. 5f). These results suggest a prominent role of sex chromosomes in the control of dietary fat absorption, the regulation of serum lipids and the development of atherosclerosis. If results from these experimental studies are translatable to humans, then thrifty effects of an XX sex chro-mosome to promote fat absorption and handling, effects that may be useful for child-bearing women, could adversely influence health when protective effects of female sex hormones are lost upon menopause (Fig. 5f).

Sex hormones exert many effects that have been suggested to contribute to sex differences in fat storage, circulating lipids and cardiovascular diseases[29]. By comparison, the role of the other primary biological determinant of sex, namely sex chromosomes, in disease development is relatively unknown despite an asso-ciation of sex chromosome abnormalities with lipid and cardio-vascular disorders[30–33]. The FCG murine model allows for determination of the relative effects of sex hormones vs. sex chromosomes to a phenotype. Recent studies using this model indicate that the dose of the X chromosome influences several metabolic traits, including obesity, fatty liver, food intake and glucose homeostasis[22,34–36]. Results from the current study extend these findings to murine models of dyslipidemia and

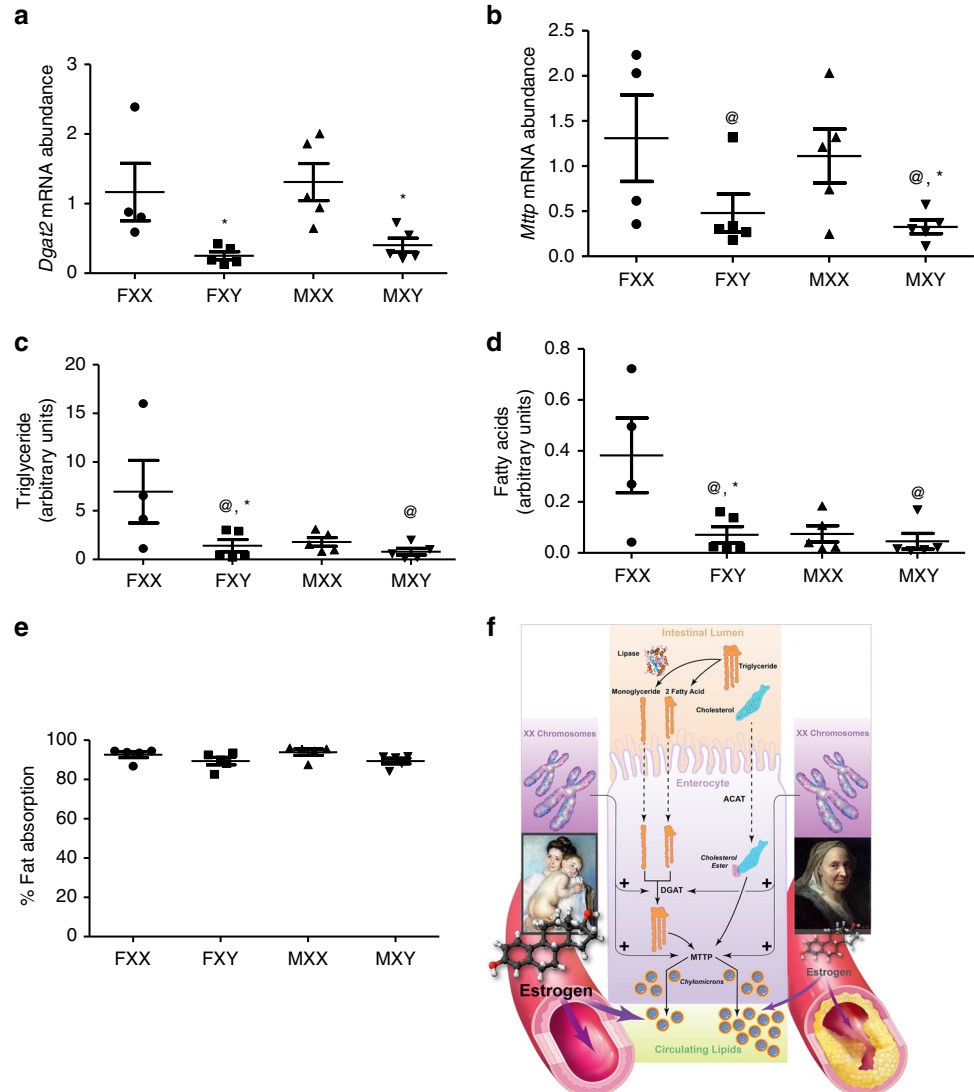

**Fig. 5** Intestinal gene expression, triglyceride content, and fat absorption are higher in XX than XY $Ldlr^{-/-}$ mice. mRNA abundance of $Dgat2$ (**a**) (FXX, $n =$ 4; FXY, $n = 5$; MXX, $n = 5$; MXY, $n = 5$) and $Mttp$ (**b**) (FXX, $n = 4$; FXY, $n = 5$; MXX, $n = 5$; MXY, $n = 5$) in small intestines from gonadectomized (GDX) male and female XX and XY mice. **c** Intestinal triglyceride content in GDX XX and XY male and female mice (FXX, $n = 4$; FXY, $n = 5$; MXX, $n = 5$; MXY, $n =$ 5). **d** Fatty acids (Palmitic/palmitic/oleic acids) that are enriched in intestinal triglycerides from XX compared to XY GDX mice (FXX, $n = 4$; FXY, $n = 5$; MXX, $n = 5$; MXY, $n = 5$). **e** Absorption of dietary fat (%) in XX and XY male and female mice. Symbols represent individual mice per group per measurement, with horizontal lines representing mean ± SEM ($n = 5$ mice/genotype/sex). *$P < 0.05$ compared to XX within gonadal sex. #$P < 0.05$ compared to female within genotype. @$P < 0.05$ XX vs. XY. **f** Overall conclusions from this study. Data (**a**–**d**) were analyzed by 2-way ANOVA with Holm–Sidak test. Dietary fat absorption data (**e**) were analyzed by Kruskal–Wallis. Source data are available as a Source Data file

atherosclerosis, demonstrating that an XX sex chromosome genotype is associated with profound elevations in pro-atherogenic circulating lipids and atherosclerosis. Moreover, these effects were observed in several different mouse models of atherosclerosis on a C57BL/6 J background ($Ldlr^{-/-}$, $Apoe^{-/-}$, wild type), with or without consumption of experimental diets, and were largely independent of sex hormones (e.g., persisted in GDX mice). These results indicate a strong differential effect of XX vs. XY chromosomes on the development of atherosclerosis.

In preclinical models and in humans, estrogen favorably influences circulating lipid profiles and atherosclerotic lesion development[7–10]. In agreement, our results demonstrate that removal of ovarian hormones increased atherosclerosis in XX mice, and extend previous results by demonstrating that this protective effect of ovarian hormones was dependent on an XX sex chromosome genotype, as XY females were not influenced by

gonadectomy. Beneficial effects of estrogens have been suggested to protect females from CAD.(reviewed in[37]) However, with advancing age (>85 years of age), the prevalence of atherosclerosis in females has been reported to not only catch up to, but in some studies exceed that of males[11–13]. Our results demonstrate that in addition to regulation of atherosclerosis by sex hormones, an XX sex chromosome genotype, relative to XY, promotes athero-sclerosis in male or female mice. Future studies should determine if an XX chromosome genotype contributes to a higher pre-valence of atherosclerosis in aged, postmenopausal females.

Recent studies demonstrated higher HDL-cholesterol levels in XX compared to XY mice of the FCG fed standard or HF diets[36]. We observed higher levels of HDL-cholesterol in XX compared to XY males, which were no longer evident in GDX males, sug-gesting an effect of testosterone or its metabolites to promote circulating HDL-cholesterol. In this study mice with an XX sex

chromosome genotype had greater concentrations of total and LDL-cholesterol, as well as markedly increased serum TG concentrations compared to XY mice, presumably related to the hyperlipidemic background of genetically manipulated ($Ldlr^{-/-}$, $Apoe^{-/-}$) mice. Levels of circulating pro-atherogenic lipids in atherosclerosis-susceptible $Ldlr^{-/-}$ mice from the present study were far in excess of those observed in previous studies using atherosclerosis-resistant C57BL/6 J mice that typically carry lipids predominately on HDL particles[36]. Moreover, multiple linear regression analysis demonstrated that genetic background, such as C57BL/6 J mice with predominately HDL- rather than LDL-cholesterol, when compared to $Apoe^{-/-}$ mice as the reference background, was an experimental variable that correlated significantly to the extent of atherosclerosis.

Previous results from the Framingham Heart Study demonstrated that increases in circulating levels of TG correlated more strongly to cardiovascular disease risk in women compared to men, but mechanisms for this sex difference have not been identified[2,38]. In this study, serum TG concentrations were higher in XX males than XX females, but atherosclerosis in the aortic sinus was similar between XX males and females, suggesting that female XX mice have more atherosclerotic burden than XX males for a given level of circulating TG. The profound effects of an XX sex chromosome genotype to promote higher levels of circulating pro-atherogenic lipids, regardless of background genetic strain, sex hormones, or diet, could result from alterations in hepatic lipid/cholesterol production and/or secretion, or alterations in intestinal lipid/lipoprotein absorption. However, surprisingly, multiple linear regression did not identify serum cholesterol concentrations as an explanatory variable for the development of atherosclerosis. Thus, while serum concentrations of pro-atherogenic lipids are clearly important in the development of atherosclerosis, in these studies they were not the primary contributing variable to the extent of atherosclerosis. In contrast, sex chromosome genotype, namely an XX sex chromosome genotype, did correlate significantly to the extent of atherosclerosis.

Notably, expression levels of a large number of genes in liver were influenced by sex chromosome genotype, with greater expression of genes involved in immune function in livers from XY compared to XX mice. Surprisingly, gene pathways implicated in cholesterol synthesis and handling by liver were not altered in XX mice despite markedly higher serum cholesterol and TG concentrations. These results are in agreement with previous findings indicating that differences in plasma lipids between XX and XY mice (male or female) were not associated with alterations in liver gene expression levels for components of cholesterol synthesis and metabolism[36]. Moreover, these results extend previous findings by demonstrating the large impact of sex chromosome genotype on hepatic gene expression patterns. Consistent with a lack of effect of sex chromosome genotype on cholesterol handling genes, hepatic TG secretion was not altered in XX male or female mice fed standard murine diet. However, future studies should more fully characterize the contribution of hepatic production of apolipoprotein B100 to the observed effects of sex chromosome genotype on the development of atherosclerosis.

Since hepatic secretion of TG did not appear to be a primary target for regulation by sex chromosome complement, we turned to intestinal handling of lipids as a cause of higher circulating lipids and cholesterol in XX mice. Small intestine gene expression of Dgat2 and Mttp, enzymes involved in the synthesis of TG from absorbed fatty acids and assembly into chylomicrons, respectively, paralleled changes in circulating lipids, with higher levels in intestines from XX than XY males or females, and these effects were independent of gonadal hormones as they persisted in tissues from GDX mice. As these genes are not X-linked and do not reside on the X chromosome, their regulation may be indirect or downstream of X chromosomes. Moreover, elevations in gene expression of Dgat2 and Mttp in small intestines from XX compared to XY mice (male or female) were accompanied by greater intestinal TG and fatty acid content, in a manner that reflected lipids within the Western diet. Recent studies identified a role for the gut microbiome in sexual dimorphism of gene expression in mice[39], sex differences in gut microbiota composition[40], and differences in the composition of gut microbiota have been demonstrated between genders and between women of different hormonal status[41]. In agreement with previous findings[40], we found that alpha diversity of gut microbiota was influenced by sex, but not necessarily by sex chromosome genotype. These results, while interesting, do not suggest a primary role for the gut microbiome in augmented fat absorption, higher serum lipids and atherosclerosis of XX compared to XY mice. Rather, absorption of dietary fat was modestly, but not significantly higher in XX compared to XY mice, indicating that altered expression levels of these pivotal lipid-regulating genes were accompanied by functional changes in fat bioavailability. The modest increase in daily fat absorption of XX mice observed in this study, when considered cumulatively over 4 months of the Western diet and in conjunction with increased energy intake, most likely contributed to the observed hyperlipidemia of XX compared to XY mice.

In conclusion, results from this study identify a profound effect of an XX sex chromosome effect to promote experimental dyslipidemias and atherosclerosis, a finding of potential relevance to increased CAD of postmenopausal females. These effects of an XX sex chromosome genotype were associated with augmented absorption and bioavailability of dietary lipid, metabolic traits that if present in humans may be important for child-bearing purposes (Fig. 5f, left). However, our findings suggest that these effects of an XX sex chromosome genotype, upon loss of protective sex hormones, may contribute to elevations of pro-atherogenic lipids and increased atherosclerotic burdens of postmenopausal females.

## Methods

**Animals**. Male mice with deletion of the gene Sry from the Y-chromosome, but expressing Sry transgene on an autosome (termed FCG mice) aged 8–12 weeks were bred to females of 3 different backgrounds ($Ldlr^{-/-}$, $Apoe^{-/-}$, and C57BL/6 J) to generate male and female mice with an XY or an XX sex chromosome complement. Mice genotypes were identified by amplifying DNA extracted from tail or ear clips using a Promega Maxwell system and polymerase chain reaction (PCR) using a commercial PCR mix (Promega 2X Master Mix, cat#m7123) and specific primers for the Sry transgene, presence of the Y-chromosome, and internal positive control. $Ldlr^{-/-}$ FCG mice (12–16 weeks old) were fed Western diet, 42% kcal from fat (TD88137, Harlan Teklad, Indianapolis, IN), $Apoe^{-/-}$ FCG mice (14–16 weeks old) were fed standard murine diet (Purina 5001; approximately 5% fat, PMI Nutrition International, St. Louis, MO), and C57BL/6 J FCG mice (14–16 weeks old) were fed an atherogenic diet (7.5% cocoa butter, 1.25% cholesterol, 0.5% sodium cholate, TD90221, Harlan Teklad, Indianapolis, IN) for 16 weeks. All experiments were approved by the animal care and use committee at the University of Kentucky and the University of California, Los Angeles and conformed to the Guide for the Care and Use of Laboratory Animals published by the NIH.

**Ovariectomy (OVX)**. Female FCG mice on the $Ldlr^{-/-}$, $Apoe^{-/-}$, and C57BL/6 J background at 8–12 weeks of age were ovariectomized under anesthesia with isoflurane (3–4% for induction and 2% for maintenance). Ophthalmic ointment (Puralube vet ointment, Dechra) was applied to the eyes of mice to prevent dryness during surgery. Before surgery, mice were subcutaneously injected, and after 12–16 h of the surgery, with 10 mg/kg flunixin for analgesia. Mice were shaved in the abdominal region at both flanks and a depilatory cream (Nair, Inc.) applied to the skin to remove hair, followed by sterilizing with povidone-iodine and ethanol. At each flank, a 1 cm incision was made to allow for locating fallopian tubes, the vascular supply to ovaries was occluded using a hemostat, and the ovaries were removed. Residual ends of fallopian tubes were ligated by cauterization. The incision was closed by suturing (5–0 black monofilament nylon suture, Ethilon 1668G) the peritoneum and clipping the skin with wound clips (Autoclip stapler),

followed by sterilization of the site with povidone-iodine. Mice recovered on a heating pad.

**Orchiectomy**. Male mice on the $Ldlr^{-/-}$, $Apoe^{-/-}$, and C57BL/6 J background at 8–12 weeks of age were orchiectomized as described previously[42]. Briefly, mice were anesthetized (isoflurane, 2–3%) and given pre and postoperative analgesic (flunixin, 2.5 mg/kg). Mice were shaved in the scrotum region and a depilatory cream was applied to remove hair, followed by sterilizing with povidone-iodine/ethanol three times. After a small incision to this region, vas deferens are collapsed using a hemostat and the testes removed. The vasculature to the area is cauterized using a high temperature fine-tip look cauterizer and the hemostat released. The surgical site was closed by wound clips and treated with povidone-iodine. For sham-surgeries, the testes are manipulated but left intact in anesthetized mice. Mice were allowed to recover for 2 weeks after surgery and to allow sufficient time to clear endogenous testicular hormones.

**16 S rRNA sequencing and measures of microbiota diversity**. Male and female XX and XY FCG $Ldlr^{-/-}$ mice ($n = 4$–6 mice/group) were fed the Western diet for 4 weeks. Following a 6 h fast, mice were anesthetized for harvest of the cecum contents. DNA was extracted from cecum contents using the PowerSoil 96-well DNA Isolation Kit (MoBio, Carlsbad, CA, USA), and 16 s rRNA sequencing was conducted by the Environmental Sample Preparation and Sequencing Facility (ESPSF) at Argonne National Laboratory and analyzed by Quantitative Insights Into Microbial Ecology (QIIME) as described previously[43,44]. To estimate alpha diversity, Operational taxonomic units (OTUs, the count of unique OTUs found in a given sample) were chosen using open reference OTU picking against the Greengenes database and diversity indices including Chao1 (species richness), Phylogenetic Diversity (PD whole tree), and Shannon (information entropy of the observed OTU abundances, to account for both richness and evenness of species) were calculated.

**Measurements of plasma and serum components**. Concentrations of total serum cholesterol, triglyceride and testosterone were quantified in sera (blood collected from cardiac puncture) using enzymatic assay kits (Total cholesterol; FUJIFILM Wako Diagnostics USA, cat#999–02601 and Triglyceride; L-type TG cat#994–02891 color A, cat# 990–02991 color B, and Alpco, cat#55-TESMS-E01; respectively). Plasma lipoprotein cholesterol was determined by on-line, high performance gel filtration chromatography using Infinity Cholesterol reagent (Thermo).

**Quantification of atherosclerosis**. Atherosclerotic lesions in the aortic arch and aortic sinus were quantified as described previously[45,46]. Briefly, cleaned aortas were cut open longitudinally and mounted on a black wax background using pins (Fine Science Tools, cat# 26002–20). Lesions, appearing as white tufts on a translucent aortic wall background, were traced and the quantification of lesion area is represented as a percent of the total intimal surface.

**Quantification of whole body metabolism**. Indirect calorimetry was performed using a LabMaster system (TSE Systems Inc., St. Louis, MO). Mice were acclimated to chambers for one week, then placed on recording platforms for five days. Data from three 24-h periods were averaged and analyzed by ANCOVA plot vs. final lean mass.

**Measurement of liver VLDL secretion**. Male and female FCG XX and XY $Ldlr^{-/-}$ mice fed standard murine diet were fasted for 4 h, anesthetized with isoflurane (4% induction & 2–3% maintenance, inhalation), and injected retro-orbitally with (1) [$^{35}$S]Met/Cys (7 µCi/g body weight; Cat #: NEG772007MC, Perkin Elmer, Waltham, MA) to radiolabel newly synthesized apoB and (2) poloxamer 407 (1,000 mg/kg, i.p.; USP grade, BASF Corporation, Florham Park, NJ) to block lipolysis. Artificial tears were applied to lubricate and protect eyes following retro-orbital injection. Blood (2 drops, about 50–70 µl) was drawn from the submandibular vein at 0 h (immediately prior to injection), 30 min, 1, 2, and 3 h post poloxamer injection. Blood was immediately centrifuged at $7,600 \times g$ for 10 min at 4 °C. Plasma was collected and used to quantify TG concentration by enzymatic assay (Wako, TG kit Cat# 461–09092 and 461–08992). TG secretion rates were derived from the slope of the line of best fit of time vs. plasma TG concentration (mg/dL) for each individual animal using GraphPad Prism 5. After the last collection time point, mice were euthanized for blood collection via heart puncture. To measure secretion of newly synthesized apolipoprotein B, plasma samples (3 h, 10 µl) were immunoprecipitated with goat anti-human apolipoprotein B antiserum (5 µg; Cat # 20S-G2, Academy Bio-Medical Company, Houston, TX) in buffer containing 1% Triton X-100, 0.1% SDS, 0.5% sodium deoxycholate, 0.2% bovine serum albumin, protease inhibitors, and phosphatase inhibitor for 18 h using rotation at 4 °C, and then 20 µl of protein G beads (50:50 slurry; GE Healthcare Amersham) were added for an additional 2 h incubation. Beads were collected by centrifugation ($9,391 \times g$ for 10 s) and washed three times with lysis buffer. Proteins were eluted from the beads by heating (70 °C for 10 min) in SDS-PAGE sample buffer and fractionated by 4–15% gradient SDS-PAGE (Catalog #

567–1085, Biorad, Hercules, CA). Gels were dried and images were acquired by autoradiography[47,48].

**Lipid measurement in intestine and liver**. Lipidomic analysis was performed using an Ultimate 3000 ultrahigh performance liquid chromatography system coupled to a Thermo Q-Exactive Orbitrap mass spectrometer equipped with a heated electrospray ion source (Thermo Scientific, CA, USA). Lipid extracts were separated on a Waters ACQUITY BEH C8 column ($2.1 \times 100$ mm, 1.7 µm) with the temperature maintained at 40 °C. The flow rate was 250 µL/min, and the mobile phases consisted of 60:40 water/acetonitrile (A), and 90:10 isopropanol/acetonitrile (B), both containing 10 mM ammonium formate and 0.1% formic acid. The samples were eluted with a linear gradient from 32 to 97% B over 25 min, maintained at 97% B for 4 min and re-equilibrated with 32% B for 6 min. The sample injection volume was 5 µL. The mass spectrometer was operated in positive ionization mode, and the full scan and fragment spectra were collected at a resolution of 70,000 and 17,500, respectively. Data analysis and lipid identification were performed using the software Lipidsearch 4.1.30 (Thermo Fisher, CA, USA). Mass labeled d13-PC (18:0) was used as an internal standard.

**Quantification of dietary fat absorption**. Male and female 12 week old FCG $Ldlr^{-/-}$ mice were housed individually and fed a butterfat 5% sucrose polybehenate diet for 4 days (ad libitum) and bedding were replaced daily. Fecal pellets (5–8/animal) were collected during the third and fourth day of diet consumption and the percentage of fat absorption was quantified by the University of Cincinnati Metabolic Phenotyping Center by measuring the ratio of fat to behenate in the fecal pellet as described previously[28]. Briefly, mice were housed individually and fed ad libitum a diet containing 5% sucrose polybehenate for 3 days. Fecal pellets were collected from the animal cage each day, and approximately 10 mg of randomly sampled feces were saponified, methylated, and extracted with 0.5 N methanolic sodium hydroxide (4 mls) in a heated water bath at 80 °C (5 min). After cooling, BF3 in methanol (3 mls) was added to methylate the sample, which was heated in the water bath (5 min). After cooling, a saturated solution of sodium chloride (2 mls) and hexane (10 mls) were added, samples were vortexed (1 min) and centrifuged to create two layers. The hexane fraction was transferred to sodium sulfate (10 mg), and then the sample (1 µl) was analyzed by gas chromatography. The absorption of fatty acids was calculated as the fraction of absorbed fat as follows: $= F_dB_d - F_tB_fF_dB_d$, where $F_d =$ sum of the masses of all dietary fatty acids, $B_d =$ mass of dietary behenic acid, $F_f =$ sum of the mass of all fecal fatty acids, and $B_f =$ mass of fecal behenic acid.

**Quantitative real-time PCR**. Following 4 months of Western diet feeding, RNA was isolated from the proximal portion of the intestines (duodenum) of gonadectomized XX and XY male and female $Ldlr^{-/-}$ mice ($n = 4$–5 mice/group) and 1 µg of RNA were reverse transcribed to complementary DNA using the qScript$^{TM}$ cDNA Supermix (Quanta Biosciences, cat# 95048–500). mRNA abundance was measured by real-time PCR using SYBER Green FastMix (Quanta Biosciences, cat# 95071–012) on a BioRad quantitative real-time PCR thermocycler. Sequences for primers for RT-PCR are in Supplementary Table 1. mRNA abundance was quantified as a fold change using the ΔΔCt method and normalized to the average of the 3 least variable housekeeping genes (beta-actin, glyceraldehyde 3-phosphate dehydrogenase, and beta-2-microglobulin).

**Liver microarray**. *DNA microarrays.* Harvested liver RNA samples from GDX male and female XX and XY FCG $Ldlr^{-/-}$ mice ($n = 4$–5 mice/group) were of sufficient quality and did not differ significantly among treatment groups (Agilent Bioanalyzer RNA Integrity Number [RIN]: $9.55 \pm 0.05 - p > 0.29$; two-way ANOVA main effect of Sex $p = 0.25$; main effect of chromosome $p = 0.13$; interaction $p = 0.424$). Extracted RNA was labeled and hybridized to Affymetrix Mouse Transcriptome Array 1.0 (MTA-1.0; one array per subject). Signal intensities were calculated using the Robust Multi-array Average (RMA) algorithm[49] at the transcript level in Genomics Suite (Partek, St Louis). Data were transferred to flat files in Excel and associated with Gene Expression Omnibus annotations for this microarray platform (accession code GSE119497). Pre-statistical filtering retained unique, annotated probe sets with signal intensity ≥ 4.2 on at least 2 arrays in the study. Filtered signal intensities were analyzed by two-way ANOVA to identify significant main effects of genotype (XX vs. XY), phenotypic sex (Male vs Female), as well as Interaction. The False Discovery Rate (FDR) procedure[50], as modified by Storey[51] was used to control for the error of multiple testing ($q \le 0.01$). The complete list of significant results is provided as supplemental information (Supplementary Table 1). Functional categorization for each expression pattern was determined with the prestatistically filtered gene list as a background using DAVID bioinformatic tools[52]. Currently, DAVID does not support Affymetrix Mouse Transcriptome Array 1.0 accession identification numbers, and therefore best match accession identification numbers from Affymetrix Mouse 1.0 Exon arrays were used, covering more than 90% of the filtered Affymetrix Mouse Transcriptome Array data set.

**Statistical analyses**. Data are presented as mean ± standard error of the mean. Data were analyzed using two-way ANOVA with between group factors of gonadal

sex and sex chromosome complement. For some studies, we performed a three-way ANOVA with between group factors of gonadal sex, sex chromosomes and surgery or diet. If data were not normally distributed, they were transformed prior to ANOVA and post hoc analysis. Kruskal–Wallis ANOVA on rank was performed when data did not pass normality after transformations. Statistical analyses were performed using SigmaPlot software (Version 13) and GraphPad Prism 5. A multiple linear regression model was fit to log-transformed aortic sinus atherosclerotic lesion area with the following main effects in the model: Body weight, Cholesterol, Gonadal Fat, Sex, Chromosome, Group, and Sex Organs. Significance was defined as $P < 0.05$.

**Reporting summary**. Further information on research design is available in the Nature Research Reporting Summary linked to this article.

## Data availability
All data are available from the corresponding author upon reasonable request. Raw microarray data (Fig. 4) are available through the Gene Expression Omnibus under the accession code GSE119497. The source data underlying Figs. 1a, 2a-d, 6d, h and 7c and Supplementary Figs. 1a and 5d are provided as a Source Data file.

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

## Acknowledgements

L.A.C. is partially supported by R01HL107326 from the National Institutes of Health Heart Lung and Blood Institute (HLBI), for research support cores from P20GM103527 and P30GM127211 from the National Institute of General Medical Sciences, and from the American Heart Association (18SFRB3390001). P.T. was supported through an MMPC from the NIH (U2C DK059630). A.M. was supported for equipment acquisition by NIH 1S100D021753, from NIH HLBI R01 HL120507. M.P. was supported by NIH K99ES028734. A.P.A. was supported by NIH R01 HD076125, R01 DK083561. K.R. was supported by NIH R01 DL083561 and P01 HL028481. We acknowledge the medical illustrations provided by Tom Dolan, MS, from the University of Kentucky College of Medicine, for illustrations within Fig. 5F.

## Author contributions

All authors contributed to and approved the results and provided comments on the manuscript. Writing: L.A.C. Study design and supervision: L.A.C., S.E.T., KR, R.T., E.B., A.M., A.P.A., A.J.L. Performing studies: Y.A., X.C., X.W., L.C., W.K., H.M.A., M.P., P.D., K.T., P.T.

## Additional information

**Competing interests:** The authors declare no competing interests.

