## [Peer Review File · Nature Communications]

Reviewers' comments:

Reviewer #1 (Remarks to the Author):

AlSiraj et al. conducted a series of studies to dissect the effects of genetic from gonadal sex (X and Y chromosomes vs sex hormones) on plasma lipids and atherosclerotic plaque formation. They found a primary effect of sex chromosomes on these outcome measures, such that XX conferred an increased risk for dyslipidemia and atherosclerotic plaque formation.

The work presented is novel and will have a significant impact in the field.

However, the presentation of the work is poor. The convoluted writing style and lack of a clear definition of what is referred to as male vs female (is this definition based on genes or gonads?) makes it difficult to digest the information. Key information, such as circulating sex hormone concentrations are missing. Better control or at least consideration of the potential impact of differences in body fat mass among different models and XX and XY counterparts somewhat diminishes the potential significance of the findings. Body fatness is a major determinant of lipid metabolism and atherosclerosis development, so it is unclear whether the observations made are secondary to increased body fat or directly genetically driven. The font size on many figures is too small to be legible.

Reviewer #2 (Remarks to the Author):

This manuscript describes experiments in mice in which the effects of XX vs XY chromosomes were experimentally separated from the effects of ovarian vs testicular hormones in their influence on plasma lipids and atherosclerosis. The interesting conclusion is that XX females had more atherogenic lipid profiles and more atherosclerosis that was largely independent of the gonads.

Specific comments:

1. There are a lot of different comparisons for each of the phenotypes: XX vs XY, intact male vs female, GDX male vs female. As a result, the results section is difficult to read because the comparisons are constantly changing. The authors might consider, within each section, focusing separately on each of the comparisons in turn.
2. Can the differences in atherosclerosis be explained entirely by the differences in lipid profiles?
3. The observation that 'newly synthesized apolipoprotein B100 levels were higher in XX than XY mice, regardless of gonadal sex' is interesting and may be worthy of inclusion in the primary manuscript rather than the supplement. It seems to go against the authors' statement in the heading of that section that 'liver lipoprotein secretion is not influenced by sex chromosome complement' and suggests that hepatic apoB secretion may be influenced by XX genotype. This issue bears more discussion.
4. The semisynthetic 5% sucrose polybehenate diet study is not compelling with regard to increased fat absorption in the XX mice. An acute olive oil oral fat gavage would be of interest and may demonstrate increased absorption in the XX mice.
5. The interaction between ovarian hormones and XX genotype is fascinating; did the lack of effect of ovariectomy in XY vs XX mice extend to gene expression in the intestine?
6. While these results are interesting, the authors should be careful in extrapolating too directly to humans.

Reviewer #3 (Remarks to the Author):

This is an interesting manuscript that explores the role of sex chromosome complement on circulating lipids and atherosclerosis. To this end the authors assessed atherosclerosis in multiple models and under different experimental paradigms in mice that allow distinction of effects due to XX or XY chromosomes, and male or female gonads. The manuscript is overall, well written and the studies are nicely performed and described. There are a few concerns, mostly of a minor nature that should be addressed.

The main concern is about the strain-dependent differences in atherosclerosis and lipid metabolism but the authors overcame the problem using different experimental paradigms (Ldlr^{-/-}, Apoe^{-/-}, wild type) in animals with the same genetic background (C57BL/6) than FCG mouse model. These paradigms in the atherosclerosis susceptible C57BL/6 genetic background are very widely employed

to study atherosclerosis with a variety of physiological and genetic interventions, but the author should care to generalize their findings with either of these models to fashion the overall picture of atherogenesis.

Regarding to the microbial environment, how the nature and composition of the diet can impact on gut flora?, this need to be addressed especially after the recognition that the metabolism of dietary components by gut flora may exert a substantial influence on metabolic outcomes.

The microbiota differs between the sexes, both in animal models and in humans. The authors should comment if the differences in gut microbiota observed between men and women might have a role in the definition of sex differences in the prevalence of metabolic and intestinal inflammatory diseases. The authors should briefly comment this topic.

Response to Reviewer #1: We appreciate the positive and constructive comments of the reviewer that have improved the revised manuscript. We respond below to specific concerns raised by the reviewer, and have revised the manuscript accordingly.

“However, the presentation of the work is poor. The convoluted writing style and lack of a clear definition of what is referred to as male vs female (is this definition based on genes or gonads) makes it difficult to digest the information.”

We have revised the manuscript to (1) clearly define male vs female based on gonads, (2) describe gonadal influences first, followed by chromosomal influences for each parameter measured. We hope that these changes make the results easier to interpret.

“Key information, such as circulating sex hormone concentrations are missing.”

We apologize, we did not originally include these measurements as we have previously reported serum estrogen and serum testosterone concentrations of the four core genotype model on the different genetic backgrounds used in these studies.¹⁻³ The Calbiotech Mouse Estradiol ELISA states “Do not use grossly lipemic specimens”. As illustrated in Figure 2C, *Ldlr*^{-/-} mice fed the Western diet for 4 months have grossly lipemic sera; thus, we were not able to quantify sera estradiol concentrations in mice chronically fed the Western diet. We have, however, reported equivalent serum estrogen concentrations between XX and XY females.² Moreover, as requested, we quantified sera testosterone concentrations (see Table). There was a significant overall effect of gonadal sex ($p = 0.005$) and surgery ($p < 0.001$), but not a significant effect of sex chromosome genotype ($p = 0.067$) on sera testosterone concentrations. Intact male mice (XX or XY) have significantly higher sera testosterone concentrations than female mice (XX or XY), and castration of males and females significantly reduced sera testosterone concentrations. We have included these data in the revised results section.

Serum testosterone concentrations (ng/ml)		
Gonadal sex	Intact	Castrated
Male:		
XX	2.89 ± 1.28 #	0.24 ± 0.04*
XY	1.65 ± 0.85 #	0.15 ± 0.06*
Female:		
XX	0.55 ± 0.08	0.24 ± 0.09*
XY	0.33 ± 0.14	0.06 ± 0.04*

Data are mean ± SEM.

*, $P < 0.05$ compared to intact within genotype; #, $P < 0.05$ compared to females.

“Better control or at least consideration of the potential impact of differences in body fat mass among different models and XX and XY counterparts somewhat diminishes the potential significance of the findings. Body fatness is a major determinant of lipid metabolism and atherosclerosis development, so it is unclear whether the observations made are secondary to increased body fat or directly genetically driven.”

The reviewer raises an important point, one that we have thought long and hard to try to discern, namely the contribution of increased body weight and fat mass of XX mice (or mice in general) to changes in serum lipids and atherosclerosis. As we are sure the reviewer can understand, it would be difficult to accurately control for this variable experimentally through

restrictions in food intake of XX mice (compared to XY) throughout 4 months of Western diet feeding.

To address this concern using data we have on hand, we consulted with a biostatistician. Using a multiple linear regression model, Dr. Thompson fit log-transformed atherosclerotic root lesion data as the response variable and the following explanatory variables in the model – body weight, serum cholesterol concentrations, gonadal fat (%), sex, sex chromosome genotype, group (background strain), and sex organs (intact versus gonadectomized). We used data on these parameters from all mice within the experimental design of studies within this manuscript. Our goal was to determine which, if any, of these variables was explanatory for the development of atherosclerosis. After adjusting for all other variables, body weight was not a significant term in the model. Similarly, neither serum cholesterol concentrations nor % gonadal fat were significant terms in the regression model, after adjusting for other parameters. However, sex chromosome genotype ($p < 0.001$) and group ($p < 0.0001$) were significant in the model as response variables for the development of atherosclerotic root lesions, after adjusting for other terms.

These data suggest that sex chromosome genotype is indeed a contributing variable to the development of atherosclerosis. The association of “group” to atherosclerosis is most likely the result of the low atherosclerotic lesion areas in C57BL/6J mice with low levels of circulating pro-atherogenic lipids. We have included this analysis in the revised manuscript, and thank the reviewer for the constructive input.

“The font size on many figures is too small to be legible.”

We have increased the font size on figures.

Response to Reviewer #2: We appreciate the positive and constructive comments of the reviewer that have improved the revised manuscript. We respond below to specific concerns raised by the reviewer, and have revised the manuscript accordingly.

1. *“There are a lot of comparisons for each of the phenotypes: XX vs XY, intact male vs female, GDX male vs female. As a result, the results section is difficult to read because the comparisons are constantly changing. The authors might consider, within each section, focusing separately on each of the comparisons in turn.”*

We agree with the reviewer, it was very challenging to draft the results section with so many different comparisons. In the revised manuscript, we have followed the same order of description for comparisons for each measured parameter, namely differences arising from biologic sex (male *versus* female), sex chromosome genotype (XX *versus* XY), or gonadectomy (intact *versus* GDX, when applicable).

2. *“Can the differences in atherosclerosis be explained entirely by the differences in lipid profiles?”*

We provide a revised version of our response to Reviewer #1. The reviewer raises an important point, namely the contribution of serum lipids to the development of atherosclerosis as accounting for higher lesion formation in XX compared to XY mice. Using data we have on hand, we consulted with a biostatistician to define the contribution and interaction of various measured parameters within our studies. Using a multiple linear regression model, Dr. Thompson fit log-transformed atherosclerotic root lesion data as the response variable and the following explanatory variables in the model – body weight, serum cholesterol concentrations, gonadal fat (%), gonadal sex, sex chromosome complement, group (background strain), and sex organs (intact *versus* gonadectomized). We used data on these parameters from all mice within the experimental design of studies within this manuscript. Our goal was to determine which, if any of these variables was explanatory for the development of atherosclerosis. After adjusting for all other variables, body weight was not a significant term in the model. Similarly, neither serum cholesterol concentrations nor % gonadal fat were significant terms in the regression model, after adjusting for other parameters. However, sex chromosome complement ($p < 0.001$) and group ($p < 0.0001$) were significant in the model as response variables for the development of atherosclerotic root lesions, after adjusting for other terms.

These data suggest that sex chromosome complement is a contributing variable to the development of atherosclerosis. Using this approach, it does not appear that differences in atherosclerosis can be explained entirely by differences in lipid profiles. However, it is likely that the significant effect of group on atherosclerosis arose from pronounced differences in serum concentrations of proatherogenic lipids across the background strains of mice within our experimental design (e.g., C57BL/6 *versus* *Apoe*^{-/-} or *Ldlr*^{-/-} mice). Moreover, it is possible that an absolute level of plasma cholesterol is required for the development of atherosclerosis, but levels above were not a primary determinant of lesion formation in these studies. We have included this analysis in the revised manuscript, and thank the reviewer for the constructive input.

3. *“The observation that ‘newly synthesized apolipoprotein B100 levels were higher in XX than XY mice, regardless of gonadal sex’ is interesting and may be worthy of inclusion in the primary manuscript rather than the supplement. It seems to go against the authors’ statement in the heading of that section that ‘liver lipoprotein secretion is not influenced*

by sex chromosome complement' and suggests that hepatic apoB secretion may be influenced by XX genotype. This issue bears more discussion."

We agree with the reviewer that higher levels of newly synthesized apoB100 in XX vs XY mice are interesting and suggest some regulation of hepatic apoB secretion by sex chromosome complement. In the revised discussion, we point out this sex chromosome effect as a potential contributor to higher sera cholesterol concentrations in XX than XY mice. However, we point out to the reviewer that these measurements were performed in mice fed standard murine diet (so that the hepatic cholesterol secretion was not at maximal capacity), and represent data on apoB100 levels in sera from n = 4 mice at one time point (3 hours) after administration of radiolabeled precursor and poloxamer, making it difficult to compare to mice fed the Western diet for 4 months. For technical reasons related to saturation of hepatic TG secretion rates, we did not quantify these parameters in XX mice fed the Western diet for 4 months.

4. *"The semisynthetic 5% sucrose polybehenate diet study is not compelling with regard to increased fat absorption in the XX mice. An acute olive oil oral fat gavage would be of interest and may demonstrate increased absorption in the XX mice."*

We recognize that data on fat absorption using the sucrose polybehenate diet were not statistically significant ($p = 0.06$) or of pronounced magnitude in comparison to marked differences in serum lipid concentrations between 4 month Western diet-fed XX and XY mice. We used this method as it has compared favorably to fat balance measurements⁴. Moreover, this proportional method (% fat absorbed from the ingested diet) has the advantage that it is not influenced by food intake differences between genotypes (e.g., higher food intake in XX than XY mice). Thus, this method suggests that from a given amount of ingested diet, the % fat absorption is higher in XX than XY mice. In addition, this method assess fat absorption over several days rather than a single meal as analogous to olive oil gavage.

We used several approaches to more fully respond to the reviewer. First, using % differences in fat absorption between XX and XY mice illustrated in Figure 5E, we calculated the cumulative difference in fat absorption that would be anticipated between XX and XY mice consuming the Western diet chronically for 4 months. We used the average daily % fat absorption of male and female XX (93.3%) versus XY (89.4%) mice. There is a 1.70% difference in fat absorption between XX and XY mice (higher in XX than XY). Considering this difference over the 120 days of consumption of the Western diet, cumulatively, XX mice would absorb 204% more fat cumulatively over this period of time. We used the average daily kcal intake for all groups of mice (16.3 kcal/day) x 120 days = 1,956 kcal ingested by mice over 4 months. As the Teklad diet had 42% kcal as fat, this represents 821.5 kcal of fat. Since the % fat absorption of XX mice over the same time period (4 months) was 204% higher than XY mice, by comparison, XX mice would consume 1,675 kcal of fat.

Second, we provide data on fasted versus non-fasted sera TG concentrations in XX vs XY females fed the Western diet for 1 week prior to measurements (Figure 1). Differences between genotypes in sera TG concentrations that were evident in non-fasted mice were no longer evident when XX and XY females were fasted prior to measurement. These data are not surprising, as we (Figure 1 of the manuscript) and others^{5,6} have reported higher food intake of XX than XY mice, indicating that higher food intake is likely a contributor to greater sera TG concentrations of XX mice. We suggest that higher food intake, coupled with higher fat

absorption from a given amount of ingested food in XX compared to XY mice, most likely

predisposes XX mice to higher sera TG concentrations.

Figure 1. Serum TG concentrations in non-fasted (left) versus fasted (right) XX and XY *Ldlr*^{-/-} females fed the Western diet for 1 (left) or 2 (right) weeks. *, P<0.05 compared to XX.

Third, in the above described XX and XY female mice fed the Western diet for another 2 weeks prior to fasting, and also administered poloxamer to prevent *in vivo* lipolysis, we quantified serum TG concentrations at time 0, 1 and 4 hours after an oral olive oil bolus (200 μ l) (Figure 2). Compared to the polybehenate sucrose method of quantifying fat absorption, this method exhibits more variability, making it difficult to detect modest differences between groups. There was no significant difference in sera TG concentrations between genotypes following an olive oil gavage. It is likely that the modest elevations of fat absorption of XX compared to XY mice (1.7 % more fat absorbed by XX compared to XY mice) were overwhelmed by the large bolus of olive oil placed in the gastrointestinal tract, illustrated by elevations of plasma TG concentrations from 30 (time 0) to over 1200 mg/dl (3 hours) after olive oil gavage. As these data did not change the original interpretation of our findings and because of the reasons described above, we did not include these data in the revised manuscript.

Figure 2. Plasma TG concentrations in female XX and XY *Ldlr*^{-/-} mice fed a Western diet for a total of 1 month prior to olive oil gavage.

Taken together, we suggest that XX mice have modest increases in fat absorption from a given quantity of ingested food that when considered chronically in mice consuming a high fat diet, coupled with higher daily energy intake of XX mice, contributes to higher serum lipid concentrations. Higher expression levels of DGAT2 and MTTP in intestine may contribute to higher fat absorption and increased sera TG and cholesterol concentrations of XX compared to XY mice.

5. *“The interaction between ovarian hormones and XX genotype is fascinating; did the lack of effect of ovariectomy in XX vs XY mice extend to gene expression in intestines?”*

We agree with the reviewer, it is indeed fascinating that the effect of ovariectomy to increase atherosclerosis in the aortic arch only occurred in XX (and not XY) females. Similarly, orchietomy-induced increases in atherosclerosis only occurred in XY (and not XX) males. Unfortunately, we did not anticipate that these effects were associated with alterations in small intestine gene expression and function. Thus, we did not collect small intestines from non-castrated groups of male or female XX and XY mice to address this interesting question. In future studies, we plan to define mechanisms for sex hormone/sex chromosome interactions on serum lipids and the development of atherosclerosis.

6. *“While these results are interesting, the authors should be careful in extrapolating too directly to humans.”*

We apologize, we were carried away by the talents of our graphic designer and our results in Figure 5F. We have included statements in the revised manuscript that these findings would need to be validated in humans.

Response to Reviewer #3: We appreciate the positive and constructive comments of the reviewer that have improved the revised manuscript. We respond below to specific concerns raised by the reviewer, and have revised the manuscript accordingly.

“The main concern is about the strain-dependent differences in atherosclerosis and lipid metabolism but the authors overcame the problem using different experimental paradigms (Ldlr^{-/-}, Apoe^{-/-}, wild type) in animals with the same genetic background (C57BL/6) than FCG mouse model. These paradigms in the atherosclerosis susceptible C57BL/6 genetic background are very widely employed to study atherosclerosis with a variety of physiological and genetic interventions, but the author should care to generalize their findings with either of these models to fashion the overall picture of atherogenesis.”

We agree with the reviewer, and have altered the discussion to reflect care in generalizing our findings to atherogenesis.

“Regarding to the microbial environment, how the nature and composition of the diet can impact on gut flora?, this need to be addressed especially after the recognition that the metabolism of dietary components by gut flora may exert a substantial influence on metabolic outcomes. The microbiota differs between the sexes, both in animal models and in humans. The authors should comment if the differences in gut microbiota observed between men and women might have a role in the definition of sex differences in the prevalence of metabolic and intestinal inflammatory diseases. The authors should briefly comment this topic.”

We agree with the reviewer, it is intriguing to define the role of the microbial environment in the observed sex chromosome differences of our studies. To respond to this concern, we fed XX and XY *Ldlr^{-/-}* females the Western diet for 1 month prior to 16S rRNA sequencing and alpha diversity measurements of cecum contents. Results demonstrate that female mice, irrespective of sex chromosome complement, exhibited higher alpha diversity (Chao1, PD whole tree and observed OTU indices) compared to males (revised Supplemental Figure 5). These results demonstrate sex differences in gut microbiota diversity, but these differences most likely arise from sex hormone, rather than sex chromosome influences. Future studies will address the role of female and male sex hormones on gut microbiota complement in relation to the development of atherosclerosis. We have included these results in the methods, results and discussion section of the revised manuscript. We thank the reviewer for this suggestion.

Reference List

- 1 Alsiraj, Y. *et al.* Sex Chromosome Complement Defines Diffuse Versus Focal Angiotensin II-Induced Aortic Pathology. *Arterioscler Thromb Vasc Biol* **38**, 143-153, doi:10.1161/ATVBAHA.117.310035 (2018).
- 2 Alsiraj, Y. *et al.* Female Mice With an XY Sex Chromosome Complement Develop Severe Angiotensin II-Induced Abdominal Aortic Aneurysms. *Circulation* **135**, 379-391, doi:10.1161/CIRCULATIONAHA.116.023789 (2017).
- 3 Manwani, B. *et al.* Sex differences in ischemic stroke sensitivity are influenced by gonadal hormones, not by sex chromosome complement. *J Cereb Blood Flow Metab* **35**, 221-229, doi:10.1038/jcbfm.2014.186 (2015).

- 4 Jandacek, R. J., Heubi, J. E. & Tso, P. A novel, noninvasive method for the measurement of intestinal fat absorption. *Gastroenterology* **127**, 139-144 (2004).
- 5 Chen, X. *et al.* Sex differences in diurnal rhythms of food intake in mice caused by gonadal hormones and complement of sex chromosomes. *Horm Behav* **75**, 55-63, doi:10.1016/j.yhbeh.2015.07.020 (2015).
- 6 Chen, X., McClusky, R., Itoh, Y., Reue, K. & Arnold, A. P. X and Y chromosome complement influence adiposity and metabolism in mice. *Endocrinology* **154**, 1092-1104, doi:10.1210/en.2012-2098 (2013).

REVIEWERS' COMMENTS:

Reviewer #1 (Remarks to the Author):

No further comments

Reviewer #2 (Remarks to the Author):

None

Reviewer #3 (Remarks to the Author):

The authors have significantly improved their manuscript and have clarified their major points and conclusions. This paper is now fully satisfactory in my opinion.